# Prediction of Second Melting Temperatures Already Observed in Pure Elements by Molecular Dynamics Simulations

**DOI:** 10.3390/ma14216509

**Published:** 2021-10-29

**Authors:** Robert F. Tournier, Michael I. Ojovan

**Affiliations:** 1UPR 3228 Centre National de la Recherche Scientifique, Laboratoire National des Champs Magnétiques Intenses, European Magnetic Field Laboratory, Institut National des Sciences Appliquées de Toulouse, Université Grenoble Alpes, F-31400 Toulouse, France; 2Department of Materials, Imperial College London, London SW7 2AZ, UK; m.ojovan@imperial.ac.uk; 3Department of Radiochemistry, Moscow State University, 119991 Moscow, Russia

**Keywords:** melting enthalpy and entropy, second melting temperature, melting entropy reduction, crystallization enthalpy reduction, undercooling, overheating, homogeneous nucleation, glasses, liquid–liquid transitions

## Abstract

A second melting temperature occurs at a temperature T_n+_ higher than T_m_ in glass-forming melts after heating them from their glassy state. The melting entropy is reduced or increased depending on the thermal history and on the presence of antibonds or bonds up to T_n+_. Recent MD simulations show full melting at T_n+_ = 1.119T_m_ for Zr, 1.126T_m_ for Ag, 1.219T_m_ for Fe and 1.354T_m_ for Cu. The non-classical homogeneous nucleation model applied to liquid elements is based on the increase of the Lindemann coefficient with the heating rate. The glass transition at T_g_ and the nucleation temperatures T_nG_ of glacial phases are successfully predicted below and above T_m_. The glass transition temperature T_g_ increases with the heating rate up to T_n+_. Melting and crystallization of glacial phases occur with entropy and enthalpy reductions. A universal law relating T_n+_ and T_nG_ around T_m_ shows that T_nG_ cannot be higher than 1.293T_m_ for T_n+_= 1.47T_m_. The enthalpies and entropies of glacial phases have singular values, corresponding to the increase of percolation thresholds with T_g_ and T_nG_ above the Scher and Zallen invariant at various heating and cooling rates. The G-phases are metastable up to T_n+_ because the antibonds are broken by homogeneous nucleation of bonds.

## 1. Introduction

Glass transition temperatures are observed at a temperature T = T_g_ during heating of quenched melts. Below T_g_, atomic bonds system produces enthalpy relaxation between the two homogeneous nucleation temperatures T_n−_ of the glassy phase with the highest being T_g_ [1,2]. The glass formation at the lowest T_n-_ occurs in hyperquenched glass-forming melts at the departure of the enthalpy relaxation [3,4,5,6]. Heating the glass through T_g_ breaks the atomic bonds and gives rise to configurons that are always accompanied by a second-order phase transition [7,8,9,10,11,12]. The non-classical homogeneous nucleation (NCHM) model predicts the temperatures of glasses, stable and ultrastable glasses [13,14,15,16,17,18,19,20,21,22,23,24,25], and glacial phases [26,27,28,29,30,31,32,33,34,35] showing that a new phase called Phase 3 appears after heating the quenched liquids through T_g_ with an enthalpy equal to the difference ∆ε_lg_ between those of liquids 1 and 2. Quenched Liquid 1 has an initial enthalpy, before giving rise to the glass state, equal to ε_ls_ H_m,_ varying with the square of the reduced temperature θ = T−T_m_)/T_m_ as shown in Equation (1) (H_m_ being the melting heat of crystals) [36]:(1)$$\varepsilon_{ls}\left(\theta \right)=\varepsilon_{ls0}\left(1-\theta^{2\;}\times \theta_{0m}^{-2}\right)$$

Liquid 2 has an enthalpy equal to ε_gs_ H_m_ in Equation (2):(2)$$\varepsilon_{gs}\left(\theta \right)=\varepsilon_{gs0}\left(1-\theta^{2}\times \theta_{0g}^{-2}\right)+∆\varepsilon$$

The reduced temperatures θ_0m_ and θ_0g_ are the Vogel–Fulcher–Tammann temperatures of these two liquids and are defined by the coefficients ε_ls0_ and ε_gs0_, depending on the nucleation temperatures in the two liquids equal to [ε_ls_ (θ) − 2]/3 for ∆ε = 0 [1,36]. The reduced glass transition temperature θ_g_ is used to define them [37,38,39]. The coefficient ∆ε only intervenes in the enthalpy of quenched liquids, stable and ultrastable glasses and glacial phases. The melting enthalpy and entropy are H_m_ and S_m_ at the melting temperature T_m_. The specific heat jump at T_g_ is H_m_ d(∆ε_lg_) / dT equal to 1.5 S_m_ in a wide fraction of glasses [40].

The existence of Phase 3 was discovered for the first time in the supercooled water [41,42] and extended to glacial phases [2,31]. It appeared later that Phase 3 was built by heating the melt from its glassy state and was the congruent configuron phase expected since many years at the percolation threshold of broken bonds at T_g_ [33,43]. The name of Phase 3 was extended to all phases having a reduced enthalpy which results by cooling from a first-order transition giving rise to an exothermic enthalpy at their nucleation temperature. The first-order character disappears during the second cooling after subsequent heating above this new temperature T_g_ and leads to the zero enthalpy of glassy phase with an increased transition temperature T_g_ during the next heating.

Consequently, the formation of stable and ultrastable glasses by vapor deposition and glacial phases by heating or annealing glass-forming melts above T_g_ induces new liquid states having higher glass transition temperatures. This new point is generally not considered in the analysis of properties attached to polyamorphism. The new glass transition determines the new liquid phases at higher temperatures.

High heating rate increases the glass transition temperature [44]. Recent studies showed two associated transitions to glacial phases for various heating rates [35]. Exothermic transitions were observed during the first heating at the nucleation temperature of glacial phases followed, at higher temperatures, by endothermic glass transitions. Such observations are also found in molecular dynamics simulation of silver and silver alloys [45,46]. These findings confirmed that the glass transition characterizes Liquids 1 and 2 and Phase 3 and any change of T_g_ induces new liquid state.

Second melting temperatures of pure elements were predicted in 2007 [36]. At this time, it was already known that crystals covered by a solid thin film or imbedded into a matrix could melt at higher temperatures than T_m_ by homogeneous nucleation in crystal hearts instead of surface melting [47,48,49,50]. This idea was relaunched in glass-forming melts accompanied by predictions of their melting temperatures T_n+_ > T_m_ using the non-classical model of homogeneous nucleation [1,33,34,51] confirmed by experimental observations [52,53,54,55,56,57,58,59,60,61,62,63,64,65,66,67,68]. These predictions had established that the medium-range order persists in liquids from T_g_ up to T_n+_ due to residual bonds producing endothermic enthalpy or due to antibonds producing exothermic enthalpy at T_n+_ where the homogeneous state of liquids appears. There are several temperatures T_n+_ in non-congruent materials associated with the various solidus and liquidus temperatures [1]. Exothermic enthalpy is always obtained after heating quenched melt far from below T_g_ or after heating stable and ultrastable glass and during formation of Phase 3 above T_g_ [33,34]. Residual and new bonds are also associated with slow heating and cooling of the melt. Annealing a melt between T_m_ and T_n+_ induces bonds and may develop bond percolation leading to crystallization at T_m_ without undercooling. The temperatures T_n+_ are not easy to recognize because they are mixed with liquidus temperatures in non-congruent materials [69,70,71].

The glassy state in liquid elements was determined with ε_ls0_ and ε_gs0_ = 0.217 corresponding to the mean value 0.103 of their Lindemann coefficient. The enthalpy coefficients ∆ε_lg0_ of Phase 3 being equal to zero at T_m_, the glass transition occurs at T_g_ during cooling through a first-order transition and an exothermic latent heat equal to ∆ε_lg_ (θ_g_) [72]. Liquid ^4^He in a confined space under pressure is amorphous and undergoes a transition during heating which looks like a first-order transition [73,74,75].

The aim of this work is to predict the melting temperatures of pure elements having a Lindemann coefficient around to 0.103 with the NCHM model [72,76]. New values of Lindemann constants for each element [77] are used to calculate the glass transition temperature of elements at low and high heating and cooling rates, the transition temperatures of glacial phases and the various melting temperatures of Ag, Cu, Zr, Ta, Re, Ni and Co compared with those deduced from MD simulations or from crystallization enthalpy of highly supercooled liquid elements. 

## 2. Thermodynamic Consequences of Bonds or Antibonds Presence above T_m_

The enthalpy recovery during heating up to T_n+_ is endothermic for bonds and exothermic for antibonds breaking [1]. The new melting entropy S depends on the reduced temperature θ_n+_ because this homogeneous nucleation temperature occurs for the enthalpy change ∆ε_lg_ (θ_n+_) [1] and S is given in Equation (3):(3)$$S=\frac{H_{m}}{T_{m}}\pm \frac{\theta_{n+}H_{m}}{T_{n+}}=\frac{H_{m}}{T_{m}}\left(1\pm \frac{\theta_{n+}}{1+\theta_{n+}}\right)$$

The melting entropy S with antibonds is weaker than S_m_ = H_m_/T_m_ and equal to H_m_/T_n+_ instead of H_m_/T_m_ while total melting heat at T_n+_ is still equal to H_m_. Two separate melting temperatures with antibonds exist: the first one at T_m_ producing an entropy equal to S_m_ and the second one at T_n+_ reducing the total entropy S. Consequently, a full melting temperature T_n+_ can exist without changing the total melting heat H_m_. This metastable phase could be a glass up to T_n+_ if the glass escapes from crystallization with high heating rates. This temperature is observed with high heating rates after the formation of glass and glacial phases at lower temperatures. A residual exothermic latent heat is still observed in some glass-forming melts with heating rates R^+^ equal to 0.33–0.66 K/s [53,54,62,63] while R^+^ ≅ 10^12^–10^13^ K/s in liquid elements leads to full melting at T_n+_ [45,78,79]. The crystallization entropy is expected to be equal to S_m_ / (1 + θ_n+_) and the crystallization enthalpy H_m_ / (1 + θ_n+_) after the formation of glacial phases inducing early crystallization at lower cooling rates. Crystallization occurs by heating and formation of a fraction of bonds equal to a new critical threshold after quenching from homogeneous liquid state above T_n+_. The melting entropy is increased when a percolation threshold of bonds is achieved at T_n+_. This increase is due to the nucleation of a bond percolation threshold by still-lower heating rates following low cooling rates below T_g_ from above T_n+_. The melting is produced when the configurons percolation occurs. The crystallization occurs at T_m_ after slow cooling from homogeneous melts when the bond number becomes equal to the percolation threshold [34,35,65,66]. It is important to note that the crystallization entropy loss below T_m_ is expected to be equal and opposite to the melting entropy observed above T_m_ at the temperature T_n+_ [1].

Weakened enthalpies and entropies are also observable for crystallization of highly undercooled glass-forming melts which are initially induced by glacial phase formation below T_m_. The cases of tantalum and rhenium droplets, crystallized during free fall are examined as a first example [80]. High magnetic fields now replace free fall towers to study nucleation phenomena. A double transition leading to cobalt crystallization via glacial phase formation revealed a reduction of crystallization enthalpy under B = 12 Tesla and B = 0 [81].

## 3. Application of NCHN Model to Liquid Elements: First-Order Glass Transition during the First Cooling and Second-Order Transition during Heating

Enthalpy coefficients of liquid states 1, 2 and 3 are given in Equations (4)–(6) with θ_0g_^2^ = 1, θ_0m_^2^ = 4/9 and ε_ls0_ = ε_gs0_ for pure liquid elements using Equations (1) and (2) [72]:

Liquid 1
(4)$$\varepsilon_{ls}=\varepsilon_{ls0}\left(1-\theta^{2}9/4\right)$$

Liquid 2
(5)$$\varepsilon_{gs}=\varepsilon_{gs0}\left(1-\theta^{2}\right)\;$$

Phase 3
(6)$$∆\varepsilon_{lg}=\varepsilon_{ls}-\varepsilon_{gs}=-1.25\times \varepsilon_{gs0}\times \theta^{2}.$$

New Lindemann coefficients δ_ls_ have been determined for many elements represented Figure 1 reprinted from [77]. The enthalpy coefficients ε_ls0_ = ε_gs0_ of each element depending on δ_ls_ are given in Equation (7) [72]:(7)$$\delta_{\mathrm{ls}}={\left(1+\varepsilon_{\mathrm{gs}0}\right)}^{0.5}-1$$

The reduced glass transition temperature θ_g_ occurs at the homogeneous nucleation temperature θ_n−_ = [ε_gs_ (θ_n−_) − 2] / 3 [1] which is given in Equation (8) equivalent to Equation (2) with ∆ε = 0, θ_n−_ = θ_g_ and θ^2^_0g_ = 1:(8)$$\varepsilon_{gs0}=\left(3\times \theta_{n-}+2-∆\varepsilon \right)/\left(1-\theta_{n-}^{2}\right)$$

The glass transition reduced temperatures are represented in Figure 2 for all Lindemann coefficients of Figure 1 together with the reduced transformation temperature θ_D_ above which the liquid is homogeneous in the absence of glacial phase melting above T_D_. This temperature T_D_ occurs at the homogeneous nucleation temperature above T_m_ given in Equation (9) for θ^2^_0m_ = 4/9 applied using Liquid 1 which has a lower enthalpy coefficient ε_ls_ than that of Liquid 2 above T_m_:(9)$$\theta_{D}=\varepsilon_{ls}\left(\theta_{D}\right)=\varepsilon_{ls0}\left(1-\theta_{D}^{2}\times \theta_{0m}^{-2}\right)$$

The new Lindemann coefficient δ_ls_ of Ag and Cu is 0.108 instead of 0.103 used in a previous publication [76]. The new Ag enthalpy coefficients of Liquids 1, 2, and 3 are represented in Figure 3 with ε_ls0_ = ε_gs0_ = 0.22766 determined with Equation (7). Line 1 is attached to Liquid 1, 2 to Liquid 2 and 3 and 4 to Phase 3.

During the first cooling from homogeneous liquid state [1], Phase 3 follows Line 3 (∆ε_lg_ = 0) down to the homogeneous nucleation temperature T_g_ = 469.34 K where a first-order transition with an exothermic latent heat ∆ε_lg_ × H_m_ = −0.10937H_m_ occurs, giving rise to a constant enthalpy below T_g_ [2] (pp. 42,43). The first heating from the glassy state follows line 4 because the formation of configurons leads to a second-order phase transition. Phase 3 in liquid elements would disappear at T_m_ in the absence of glacial phase. Recent MD simulations show the first-order character of all glacial phase transitions occurring during the first cooling which is confirmed with the NCHN model [45,76]. The various enthalpy reductions are cumulated during the cooling rate and are maintained during heating up to the glass transition temperature of phases. These various events are predicted using the homogeneous nucleation temperatures associated with liquids 1, 2 and 3. The latent heat of glass-phase formation is progressively recovered by heating between T_g_ and T_m_.

The main conclusions of this chapter are the first-order character of the transition from liquids state to the glassy state and the second-order phase transition from the glassy state to the ordered Liquid 3 along Line 4 in Figure 3. This point was not raised in previous publications considering that the first-order character was reversible [72,75,76].

## 4. Singular Values of Enthalpy of Glacial Phases in Liquid Elements

Glacial phases enthalpy in glass-forming melts have singular values after the first cooling: 0 for the current glass state, −H_m_ ∆ε_lg0_ = −H_m_ (ε_ls0_ − ε_gs0_), −H_m_ ∆ε_lg0_/2, H_m_ ∆ε_lg_ (θ_0m_), and −H_m_ giving rise to crystals. The nil value could correspond to equal numbers of bonds and antibonds at the percolation threshold of bonds at T_g_. The values −∆ε_lg0_ and −∆ε_lg0_/2 were involved in the formation of stable and ultrastable glasses and gave rise to zero enthalpy and an increase of T_g_ during the second cooling [1,2,33,34]. The value ∆ε_lg_ (θ_0m_) defined the glacial phase enthalpy of Mg_69_Zn_27_Yb_4_ leading to quasi-crystalline phase of same enthalpy [32,33]. These singular values could correspond to higher percolation thresholds leading to higher T_g_ and to the glassy phases or metastable crystalline phases already analyzed for various ice amorphous phases [82,83].

Singular enthalpies of glacial phases also exist in liquid elements as already shown [76]. The nil enthalpy has disappeared except in the homogeneous liquid state above T_n+_. This homogeneous liquid state can be prolongated by rapid cooling down to T_m_. Phase 3 has a negative enthalpy equal to the latent heat of vitrification of various elements at T_g_. The other enthalpy coefficients are−ε_ls0_ = −ε_gs0_, the minimum value −1.25 ε_ls0_ at 0 K, and ∆ε_lg_ (θ_0m_). Figure 4 shows these values for Ag. Table 1 gives the singular values of enthalpy coefficients for each element given in Figure 1.

## 5. Universal Law for the Second Melting Temperature T_n+_ Depending on the Homogeneous Nucleation Temperature T_nG_ of Glacial Phases

The glass transition temperature T_g_ increases with the Lindemann coefficient in Figure 2. Increasing the heating rate rises the T_g_. The NCHM model uses an increased Lindemann coefficient to predict T_g_ and T_nG_ at higher heating rates. Consequently, to each value of θ_g_ corresponds a value of ε_ls0_ = ε_gs0_ given by Equation (8) with ∆ε = 0 and new laws for ε_ls_ (θ), ε_gs_ (θ) and ∆ε_lg_ (θ) are established preserving θ_0g_^2^ = 1 and θ_0m_^2^ = 4/9. At the homogeneous nucleation reduced temperature θ_n**G**_ of glacial phase, a difference of enthalpy ∆ε_lg_ (θ_nG_) = −θ_n+_ must be induced to obtain its melting at θ_n+_ [1]. The values of ε_gs0_ = ε_ls0_ are higher than 2 in Table 2 because the melting temperature T_m_ is used instead of T_n+_ to predict T_n+_. Using the new melting temperature T_n+_, these coefficients would not be higher than 2.

Equations (4)–(8) are applied at a second melting temperature with θ_g_ = θ_n+_ for several elements as shown in Table 2. The θ_n+_ values are experimental values applying high heating rates in MD simulations [45,78,79] or observing early crystallization during droplet free fall [80] or under high magnetic field [81]. The reduced temperature θ_g_ = θ_n+_ is the highest glass transition temperature in a liquid having a reduced melting temperature equal to θ_n+_. The reduced melting temperatures θ_n+_ are chosen equal to singular values of enthalpy coefficients of various elements in agreement with those of Table 1 respecting the nucleation law ∆ε_lg_ = θ_n+_ = θ_g_ [1,36]. This calculation is extended to higher values of θ_g_ = θ_n+_ up to 1. The highest melting temperature for the highest heating rate is 2 T_m_ with a total entropy S_m_/2. For this the enthalpy ∆ε_lg_ is equal to zero and the number of antibonds would be equal to that of bonds. The total enthalpy for two separated melting temperatures at T_m_ and 2 T_m_ would be H_m_/2.

There is a maximum nucleation temperature of glacial phase equal to 1.2931 T_m_ even if the existence of much higher glass transition temperatures θ_g_ = θ_n+_ above 1.47 T_m_ is possible as shown in Figure 5. The value of θ_n+_ for Cu is much higher than that given for Ag in Table 1. The coefficient 0.35413 is the sum of 0.22766 and 0.12648. This enhancement is expected for a double transition above T_m_. It is difficult to envisage a full melting temperature far above θ_n+_ = 0.47 because triple transitions above T_m_ would be involved. The transition at θ_n+_ = 1 (not represented in Figure 5) exists because ∆ε_lg_ could be equal to the singular value ∆ε_lg_ = −1.

## 6. Observations of Second Melting Temperatures T_n+_ with MD Simulations

### 6.1. Zirconium

The nucleation temperature of Zr glassy phase occurred at T_g_ = 1000 K during the first cooling as shown in Figure 6 by [79] and predicted in Chapter 7. A second melting temperature at T_n+_ = 2378 K instead of T_m_ = 2125 K was obtained during heating after quenching the liquid below T_g_ and applying a mean heating rate R = +10^12^ K/s. The glass transition temperature T_g_ of glacial phase was equal to T_n+_ as shown in Figure 6.

### 6.2. Silver

Figure 7 published by [45] represented a sharp transition during heating at T_n+_ = 1391 K of liquid silver, corresponding to a liquid-glass transition at T_n+_ The nucleation temperature of glacial phase occurred at T_nG_ = 957 K during the heating as predicted in Chapter 7. The glass transition of Liquids 1 and 2 was T_m_ = 1234.9 K while that of glacial phase was equal to T_n+_ corresponding to θ_n+_ (θ_nG_) = 0.12468. A very high enthalpy change resulting from the first-order transitions during previous cooling was observed before undergoing the nucleation temperature T_nG_ of the glacial phase.

In Figure 8, the transition at T_nG_ = 917 K with ε_ls0_ = ε_gs0_ = 1.53040 induces a liquid phase having an enthalpy coefficient equal to −1 instead of −0.12648 and a melting temperature T_n+_ = 1391 K. The enthalpy coefficient varies from −1 to zero at T_n+_ = 1391 K.

### 6.3. Iron and Copper

The glassy phases occurred at 800 K for Cu and 1100 K for Fe as reproduced in Figure 9 and predicted in Chapter 7 [78]. Second melting temperatures T_n+_ = 1839 K instead of T_m_ = 1358 K for Cu and T_n+_ = 2207 K instead of T_m_ = 1811 K for Fe were observed, applying heating and cooling rates R = ±10^13^ K/s. The glass transition temperatures T_g_ of glacial phases were equal to T_n+_.

## 7. Glacial Phases Formation below T_m_ with Singular Enthalpies with MD Simulations

### 7.1. Silver

The steady-state relaxation time is given by Equation (10)
(10)$$ln\left(\tau_{1}/s\right)=B_{1}/(T-T_{m}/3)+ln\left(\tau_{0}/s\right)$$
with B_1_ = 1171.35 K, T_m_/3 = 411.63 K and ln (τ_0_ (s)) = −25.714 and where it was used to predict in Table 3 the glacial phase transition temperatures θ_nG_ of liquid silver during various heating and cooling rates referring to a system of 256,000 atoms reproduced in Figure 10 for few cooling rates R [45]. The value Ln(R) is used to determine T = T_g_ with Equation (10).

The NCHN model shows that weaker cooling rates induce higher nucleation temperatures of glacial phases with singular enthalpy values in agreement with MD simulations of An Q. et al. in Figure 10. A new singular value −0.15810 is added in Table 3 (Line 8) for the transition at T_nG_ = 806 K. It is equal to the difference between the singular values −0.28458 and −0.12648 given in Table 1, corresponding to the reduction of the enthalpy coefficient of Phase 3 from θ = θ_0m_ to its minimum value at θ = −1 in Figure 4. Other singular values are expected with T_g_ = T_m_ in Table 4 (Lines 14–16), −0.10937 (T_n+_ = 1370 K), −0.12648 (T_n+_ = 1391 K) and −0.15810 (T_n+_ = 1460 K), varying the heating rate. The temperatures T_nG_ are equal to 977 K (Line 14), and 957 K (Line 15) in agreement with MD simulations in Figure 7 [45]. The coefficient −0.15810 (Line 16) would lead to T_nG_ = 852 K.

### 7.2. Tantalum

A glass transition temperature T_g_ of 1650 K of tantalum was observed through ultrafast liquid quenching [84]. This value is used in Table 4 (Lines 4 and 5) to predict two possible values T_nG_ = 1980 K and 1847 K corresponding to the singular enthalpy coefficients −0.13305 and −0.16138 respectively. The same coefficients were obtained for T_g_ = T_m_ (Lines 1 and 2).

### 7.3. Zirconium

Zr MD simulations during various quenching processes revealed two T_g_ values 1000 K and 890 K and a full melting at 2378 K [79]. The same singular coefficient −0.11896 leads to T_nG_ = 1258 and 1072 K as shown Lines 7 and 9 in Table 4 and full melting occurs at T_n+_ = 2378 K. For T_g_ = 1000 K (Line 8), T_nG_ can also be equal to 1184 K with an enthalpy coefficient of −0.14009 and T_n+_ = 2423 K. For T_g_ = T_m_ (Lines 5 and 6), the two singular coefficients would be −0.11895 and −0.14009 with T_nG_ = 1661 and 1622 K leading to full melting at T_n+_ = 2378 and 2423 K, respectively.

### 7.4. Nickel

Ni MD simulations revealed two T_g_ values 1150 and 930 K [85,86]. They result from the same singular enthalpy coefficient −0.11202, Lines 12 and 13 in Table 4, which would lead to T_nG_ = 1240 and 1143 K and to full melting at T_n+_ = 1922 K. A glass transition at T_g_ = T_m_ would lead to T_n+_ = 1922 and 1953 K with singular enthalpy coefficients equal to −0.11202 and −0.13018, respectively (Lines 10 and 11 in Table 4).

### 7.5. Copper

Cu MD simulations revealed a T_g_ value of 800 K in Figure 9 with a cooling rate of −10^13^ K/s [78]. It corresponds to T_nG_ = 908 K with a singular enthalpy coefficient ∆ε_lg_ (θ_n+_) = −0.12648 (Line 18 Table 4). For T_g_ = T_m_ (Line 17 Table 4), T_nG_ = 948 K, T_n+_ = 1667 K.

### 7.6. Iron

Fe MD simulations revealed T_g_= 1000 K in Figure 9 with a cooling rate of −10^13^ K/s [78]. It corresponds to T_nG_ = 975 K with a singular enthalpy coefficient ∆ε_lg_ (θ_n+_) = −0.21882 (Line 20 Table 4). For T_g_ = T_m_ (Line 19 Table 4), T_nG_ = 1275 K, ∆ε_lg_ (θ_n+_) = −0.21882 and T_n+_ = 2207 K.

## 8. The Free Fall Solidification of Tantalum and Rhenium Droplets via Glacial Phases

The Ta and Re solidifications were observed in two steps during the free fall of overheated liquid droplets as shown in Figure 11 [80]. The undercooling (∆T) down to 2770 K was 518 K with T_m_ = 3288 K and the crystallization occurred with two steps, the first one being the first-order transition of a glacial phase at 2770 K, inducing in the second step at 2930 K full crystallization and coalescence after an incubation time. For an adiabatic process, the coalescence led to the melting temperature T_m_ with a latent heat equal to (∆T) × C_p_ = 518 × C_p_ [87], C_p_ being the average heat capacity of tantalum between 2770 and 3288 K given by Equation (11) [88]:(11)$$∆C_{p}=25+5.87\times 10^{-3}T\;\mathrm{in}\;J/K/\mathrm{mole}$$

Its average value for 2770 < T < 3288 K was 42.83 J/K/mole and the solidification enthalpy 22186 J/mole representing 69.33% of the melting enthalpy 32 KJ/mole at T_m_. The experimental reduction was 0.3067 while the sum of two singular values 0.16139 + 0.10294 = 0.29444 are predicted in Table 2. The agreement is good and θ_n+_/(1 + θ_n+_) = 0.29444 leads to θ_n+_ = 0.4424.

The Re solidification was observed at 2620 K for T_m_ = 3458 K with undercooling (∆T) = 838 K. The heat capacity was given [89] by Equation (12):(12)$$∆C_{p}=25+3.329\times 10^{-3}T\;\mathrm{in}\;J/K/\mathrm{mole}$$

The average heat capacity being 35.117 J/K/mole between 2620 and 3458 K, the solidification enthalpy was 29428 J/mole instead of 33230 Joules/mole for the melting enthalpy at T_m_. The reduction was 11.44% corresponding to the singular coefficient of 0.11981 in Table 1. The temperature T_n+_ would be equal to 1.1361 T_m_ = 3929 K. The heat capacity depends on the sample purity because a weaker value $2.29\times 10^{-3}T$ had been measured at low temperatures [88]. Its average value for 2620 < T < 3458K was 31.96 J/K/mole. The enthalpy change was 31.96 × 838 = 26782 J/mole instead of 33230 J/mole corresponding to a total reduction of 19.4%. This reduction coefficient is too far from the singular coefficients of rhenium in Table 1.

## 9. The Two Peaks of Recalescence of Undercooled Cobalt in High Magnetic Field B = 12 Tesla and B = 0

The undercooling (∆T) of liquid cobalt was increased after 20 cycles of temperature between 1873 K and 1073 K at 1 K/s [81]. These cycles tended to melt surviving nuclei due to cumulated times of annealing at 1873 K equal to the cycle number multiplied by 300 s [90]. The maximum undercooling was (∆T) = 328 ± 6 K. The annealing temperature was too weak compared to T_n+_ = T_m_ × (1 + θ_n+_). Many cycling cannot fully replace an annealing above T_n+_ because there is a first-order phase transition due to the formation of colloids [1]. The minimum value of θ_n+_ is 0.0805 in Table 1 and T_n+_ = 1910 K is 37 K higher than the annealing temperature 1873 K. The specific heat C_p_ of Co crystals is constant from 1440 K to T_m_ = 1768 K and equal to 40.6 J/K/mole [91] (p. 60). The solid fraction formed by recalescence was proportional to the temperature rising and the maximum undercooling was expected to be (∆T) = H_m_/C_p_ = 399 K instead of 328 K [87]. All the melt was crystallized here. The missing enthalpy represented a fraction equal to 0.178 ± 0.015 of that of melting 16190 J/mole. The reduced temperature θ_n+_ was equal to 0.21655 corresponding to the singular coefficients sum (0.14490 + 0.07368) given in Table 2 for Co. The temperature rising after crystallization at 1440 K, and being much weaker than T_m_, was not suitable because it was due to non-adiabaticity of the processing [81]. Two peaks of recalescence characterized by temperature rising (∆T) were observed. The first one in Figure 12 would correspond to the first glacial phase formation through a first-order transition inducing the second peak of full crystallization after an incubation time. The first peak corresponded to two values of (∆T) = 23 and 91 K for B = 12 Tesla and B = 0 representing 5.76 and 22.8% of the melting heat. The values of θ_n+_ were 0.0623 instead of 0.0805 and 0.26147 in good agreement with 0.26163 = 0.0805 + 0.18113 in Table 1.

The conclusions of this chapter are: (i) the full crystallization of liquid elements can occur with reduced entropy and enthalpy at temperatures weaker than T_m_ in glass-forming melts after heating the material from the glassy state or in the presence of a glacial phase resulting from a high undercooling rate (ii) enthalpy and entropy reductions are also expected for crystallization of quenched glass-forming melts at the well-known temperature called T_X_ < T_m_ [92] which could reveal the existence of a second melting temperature at T_n+_. This crystallization at T_X_ results from the initial formation of glacial phases by homogeneous nucleation in one or few steps. Two crystallization processes have already been observed above T_g_ in salol and triphenylethene leading to the same crystal phase with the first one attributed to homogeneous-nucleation-based crystallization at temperatures weaker than T_m_ [93,94]. Early crystallization following high undercooling shows that the enthalpies of crystallization and melting are reduced. Experiments above T_m_ to detect the value of T_n+_ and the recovered enthalpy and entropy at this temperature after early crystallization below T_m_ are necessary to confirm these proposals.

## 10. Thermodynamics of Configurons

Configurons are broken chemical bonds in condensed materials [7,8,9,10,11,12]. In crystalline materials, the configurons are highly mobile and their condensation causes the arrest of temperature at the melting point T_m_ whereas, in amorphous substances, they move with difficulties and therefore the glass-liquid transition occurs at the glass transition temperature T_g_ typically as a second order phase transformation. Following the configuron percolation theory (CPT) [7,8,9,10,11,12], the melting of a material occurs when the percolation via configurons occurs. Therefore, the melting temperature of a material (either T_g_ for amorphous or T_m_ for crystalline substances) is:(13)$$T_{m}=H_{d}/(S_{d}+RLn\left[\left(1-f_{c}\right)/f_{c}\right])$$
where H_d_ is the enthalpy and S_d_ is the entropy of formation of configurons, R is the absolute gas constant and f_c_ is the percolation threshold which is taken at low heating rates as the Scher and Zallen invariant f_c_ = 0.15 [7,8,9,10,11,12]. The second melting temperature T_n+_ that the CPT treats as percolation via unbroken chemical bonds is:(14)$$T_{n+}=H_{d}/(S_{d}-RLn\left[\left(1-f_{c})/f_{c}\right)\right])$$

Taking the ratios
(15)$$\frac{T_{n+}}{T_{m}}=\{S_{d}+RLn\left[\frac{1-f_{c}}{f_{c}}\right]/\{S_{d}-RLn\left[\frac{1-f_{c}}{f_{c}}\right]\}$$
we can find the entropies of configurons in metallic elements as
(16)$$S_{d}=RLn\left[\frac{1-f_{c}}{fc}\right]\left[\frac{T_{n+}/T_{m}+1}{T_{n+}/T_{m}-1}\right]$$


Thereafter we can calculate the enthalpies of elements as
(17)$$H_{d}=T_{m}\left(S_{d}+RLn\left[(1-f_{c})/f_{c}\right]\right).$$


Table 5 gives numerical data of configuron entropies and enthalpies and enthalpies H and entropies S of melting as calculated, decreasing with T_n+_/T_m_.

## 11. A New Panorama for Melting and Solidification

A melting heat H_m_ is measured by melting a solid element at T_m_. The melting entropy is S_m_ = H_m_/T_m_. Various thermal histories can lead to at least three liquid states: the first one has a complementary exothermic melting temperature at T_n+_, the second one has a complementary endothermic melting temperature at T_n+_ and the third one is homogeneous above T_n+_. The reduced temperature θ_n+_ is multiple and equal to singular values of enthalpy coefficients which are predicted by glacial phase formation leading to more numerous liquid states having various enthalpy differences with that of homogeneous liquid above θ_n+_. High heating rates rise the temperature of full melting up to T_n+_ which can attain and even exceed 1.47 T_m_ for liquids having high enthalpy excesses. The liquid states can be more than three depending in the number of singular enthalpies of glacial phases.

These phenomena of new liquid formation are characterized by variations of the enthalpy H_m_ and the entropy S_m_ accompanying melting and crystallization. The melting temperature rising is not only due to high heating rates and is observable in highly undercooled systems close to crystallization. Melting temperature increase at high heating rate observed in MD simulations cannot be due to overheating of the solid phase because the classical nucleation equation predicts a full melting at T_m_ without residual crystals in the melt [95]. Our description is based on the melting of superheated glassy phase at the temperature T_n+_ having an enthalpy of melting equal to that of crystalline phase. 

The existence of temperatures T_n+_ in glass-forming melts was recently confirmed at heating rates of about 0.5 K/s [1,51,52,53,54,55,56,57,58,59,60,61,62,63,64,65,66,67,68]. They are due to the formation of antibonds increasing the liquid enthalpy above T_m_ up to T_n+_ or bonds decreasing the liquid enthalpy in the interval (T_m_,T_n+_). Recent review confirms the existence of liquid-liquid structure transitions in melts above T_m_ and their impact on the following microstructure and properties after solidification [96]. A liquid-liquid transition is observed at 1575 ± 6 K in Co_81.5_B_18.5_ eutectics above a melting temperature T_m_ = 1406 K which could correspond to T_g_ = 869 ± 19 K in this fragile liquid [97]. 

The liquid microstructure is changed by cooling the melt through T_n+_ which is induced by a first-order transition from homogeneous liquid to various colloids of various compositions [54,66,68]. The colloids result from the formation of melted superatoms containing magic atom numbers discontinuously varying shell by shell with temperature [98,99]. At the temperature T_n+_, the Gibbs free energy change during cooling, favorizing the colloids formation [1], is higher than that of a continuous variation, enhanced by nucleation of bonds versus time at temperatures weaker than T_n+_, building clusters-bound colloids. These colloids contain more and more bonds and give rise to crystallization at T_m_ at low cooling rate or to glassy phase after quenching to escape from crystallization. These formations of colloids of various compositions were observed many years ago for eutectic and off-eutectic compositions [54,100,101]. 

## 12. Conclusions

Recent molecular dynamics simulations observed second melting temperatures T_n+_ of pure elements varying from 1.11896 T_m_ for zirconium to 1.35413 T_m_ for copper applying very high heating rates. The non-classical model of homogeneous nucleation (NCHN model) predicted these temperatures as melting temperatures of glassy glacial phases formed by homogeneous nucleation at temperatures T_nG_ weaker than T_n+_. These nucleation temperatures T_nG_ cannot be higher than 1.2931 T_m_. A universal law of T_n+_/T_m_ versus (T_nG_/T_m_) was obtained for all liquid elements with a Lindemann coefficient weaker than 0.137.

These glacial phases had singular constant values of enthalpy up to their upper glass transitions equal to T_n+_. The NCHN model needed singular values of enthalpy to be applied at well-defined temperatures. They corresponded to singular values of the enthalpy coefficient ∆ε_lg_ (θ) of a new phase called Phase 3 or configuron phase. These percolation threshold values must be singular to represent various organizations of elementary bricks in a glass. 

The NCHN model agreed with existing MD simulations using recent values of the Lindemann coefficient of elements presented in the columns of Figure 1 to predict the value of T_g_. Consequently, this figure and the universal law could be used to control MD simulations of other elements.

Phase 3 was induced by a first-order transition at T_nG_ by cooling and was accompanied by a second-order phase transition during reheating as predicted by formation of percolation threshold of broken bonds named configurons. 

Second melting temperatures T_n+_ have been already observed in several glass-forming melts accompanied by exothermic or endothermic latent heats at lower heating rates. They corresponded to the melting of antibonds or bonds induced by thermal history depending on cooling and heating rates from the homogeneous liquid state down to the glassy phase and from the glassy phase to high temperatures. 

The existence of temperatures T_n+_ in liquid elements has for consequence entropy and enthalpy reductions for melting. These effects are observed in all situations already described. They need to be confirmed by early crystallization of highly undercooled melts having previously undergone a glacial phase transition. We have looked at two experiments: (i) the free fall solidification of Tantalum and Rhenium droplets with two recalescence peaks and (ii) those of undercooled cobalt in a high magnetic field B = 12 Tesla and B = 0. These experiments confirmed these reductions. Observations of T_n+_ above T_m_ are expected after observing these crystallizations at T_X_ below T_m_. 

The NCHN and configuron models successfully predicted liquid–glass and liquid–liquid transformations in multicomponent glass-forming melts. These predictions are possible when the glass transition temperature T_g_ and the melting temperatures T_m_ are sufficiently precise to be able to determine the singular values of Phase 3 enthalpy. The universal law relating the nucleation temperatures T_nG_ of glacial phases and the second melting temperatures T_n+_ in multicomponent alloys could be the same to that of pure elements. 

## Figures and Tables

**Figure 1 materials-14-06509-f001:**
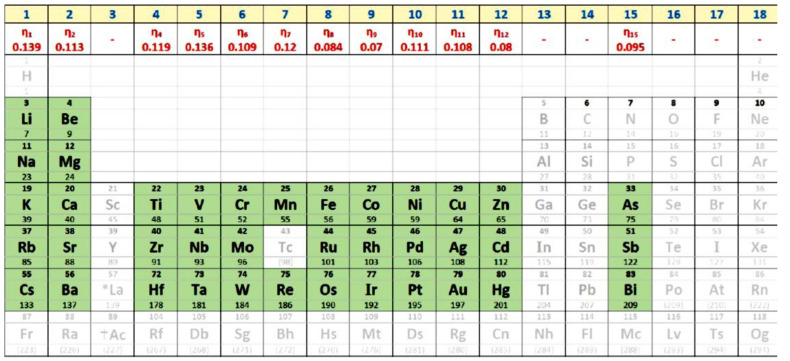
Lindemann coefficients of liquid elements distributed along columns of periodic table of elements by Mendeleev. The Lindemann coefficients δ_ls_ along Line 2 for each column. Reprinted with permission from [77]. Copyright 2019 Elsevier.

**Figure 2 materials-14-06509-f002:**
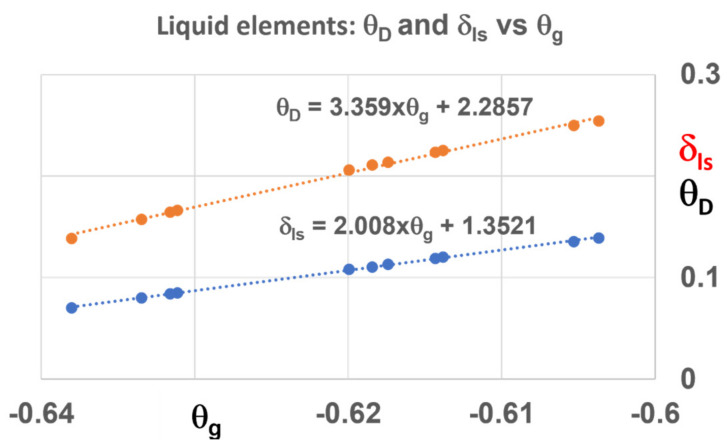
The Lindemann coefficient δ_ls_ and the reduced homogeneous nucleation temperature θ_D_ = (T_D_ − T_m_)/T_m_ of liquid elements in the absence of overheated Phase 3 above T_D_ versus the reduced glass transition temperature θ_g_ = (T_g_ − T_m_)/T_m_.

**Figure 3 materials-14-06509-f003:**
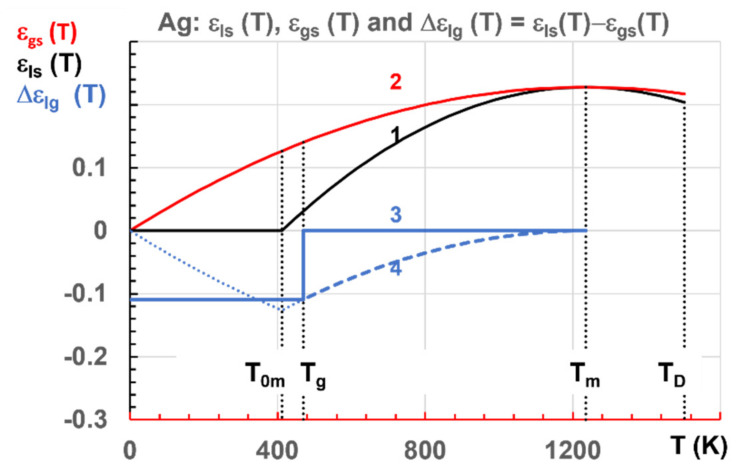
Ag enthalpy coefficients: liquids 1, 2 and 3–4 versus temperature in Kelvins. T_g_ = 469.34 K, T_0m_ = T_m_/3 = 411.63 K, T_D_ = 1489 K, ε_ls0_ = ε_gs0_ = 0.22766, θ^2^_0m_ = 0.444445 and θ^2^_0g_ = 1 in Equations (4)–(6).

**Figure 4 materials-14-06509-f004:**
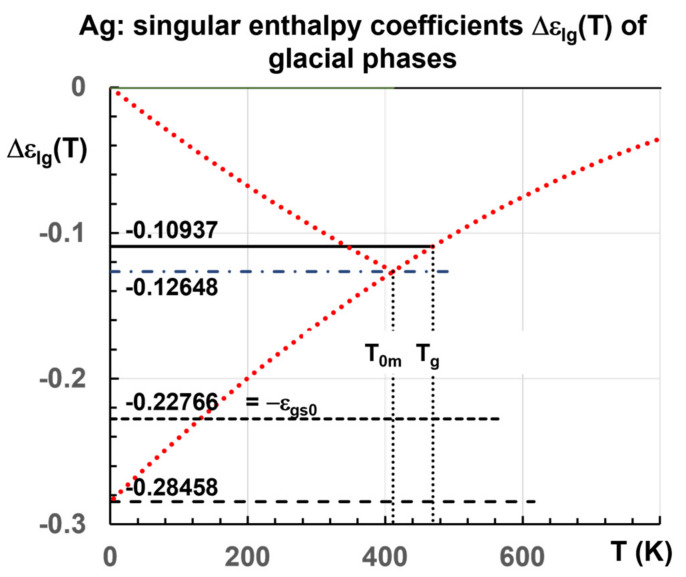
Enthalpy coefficients of Ag Phase 3 versus Temperature in Kelvins with singular values.

**Figure 5 materials-14-06509-f005:**
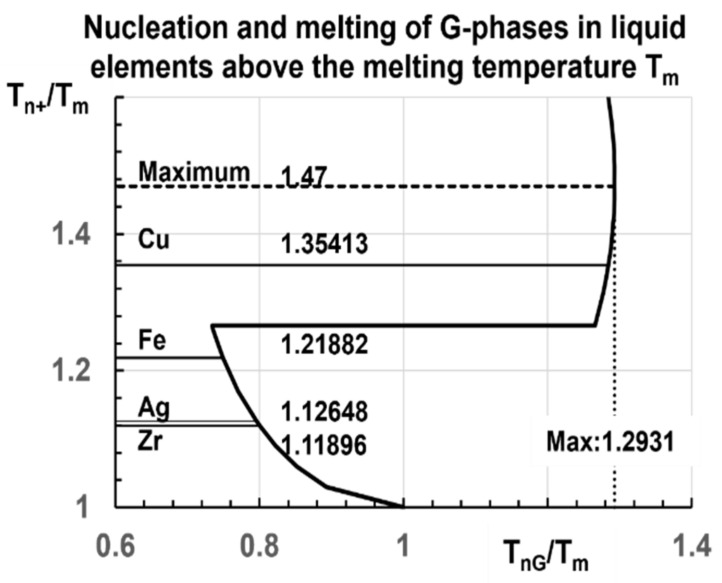
Universal diagram of T_n+_/T_m_ versus T_nG_/T_m_ of all liquid elements with higher and higher heating rates. The ratios T_n+_/T_m_ = 1.11896 for Zr (T_n+_ = 2378 K, T_m_ = 2125 K) [79], 1.12648 for Ag (T_n+_ = 1391 K, T_m_ = 1234.9 K) [45], 1.21882 for Fe (T_n+_ = 2207 K, T_m_ = 1811 K) [78], and 1.35413 for Cu (T_n+_ = 1839 K, T_m_ = 1358 K) [78] had been already observed by MD simulations at various high heating rates as shown in Figure 5, Figure 6 and Figure 7.

**Figure 6 materials-14-06509-f006:**
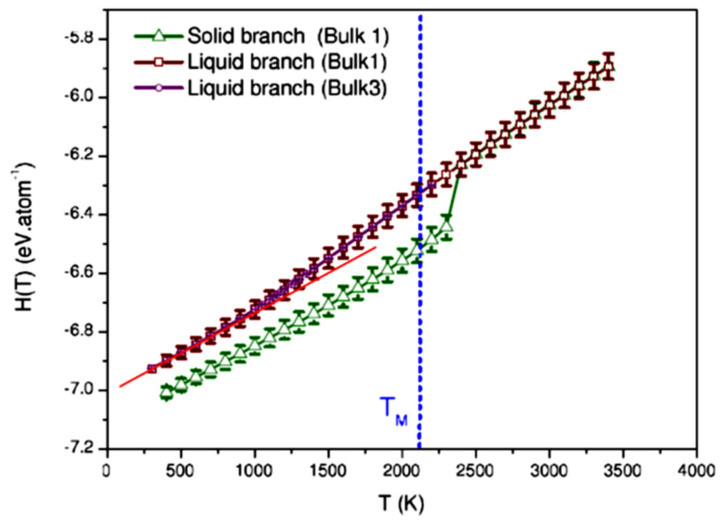
MD simulations of liquid zirconium enthalpy observing a melting temperature of 2378 K instead of T_m_ = 2125 K. Reprinted with permission from [79]. Copyright American Physical Society 2020.

**Figure 7 materials-14-06509-f007:**
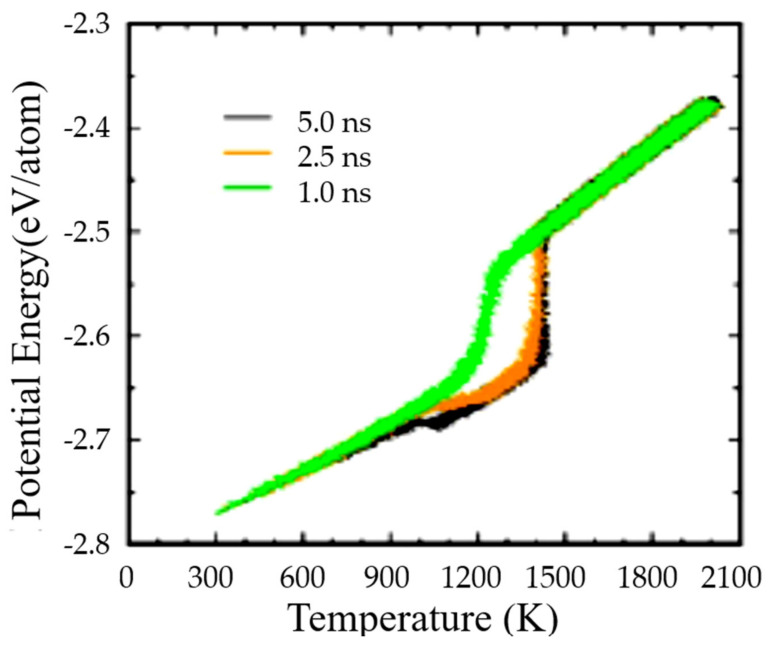
MD simulations of liquid silver observing a melting temperature of 1391 K during heating a 32,000 atoms system instead of T_m_ = 1234.9 K (5 ns with R = 0.34 × 10^12^ K/s, 2.5 ns with R = 0.68 × 10^12^ K/s and 1 ns with R = 1.7 × 10^12^ K/s). Reprinted with permission from [45]. Copyright 2020 American Chemistry Society ACS.

**Figure 8 materials-14-06509-f008:**
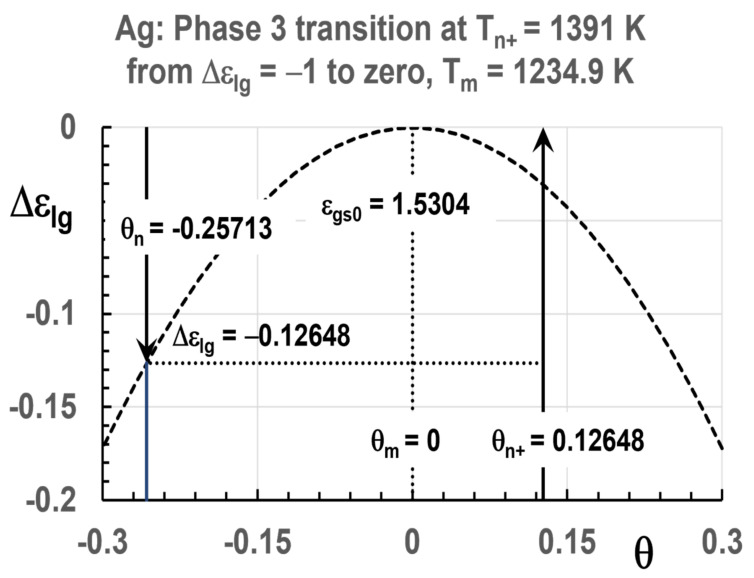
Ag Phase 3 enthalpy coefficient ∆ε_lg_ (θ) = ε_ls_ − ε_gs_ with ε_ls0_ = ε_gs0_ = 1.53040 plotted versus θ. θ_nG_ = −0.25713, ∆ε_lg_ (θ_nG_) = −0.12648 inducing ∆ε_lg_ = −1 due to the proximity of T_m_. Melting at T_n+_ = 1391 K.

**Figure 9 materials-14-06509-f009:**
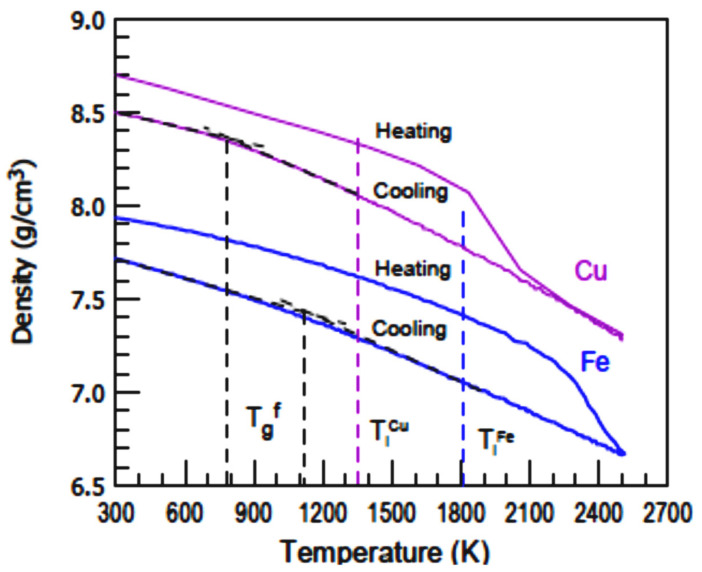
MD simulations of liquid copper density observing a melting temperature of 1839 K instead of T_m_ = 1358 K and MD simulations of liquid iron density observing a melting temperature of 2207 K instead of T_m_ = 1811 K. R = ±10^13^ K/s. Reprinted with permission from Ref. [78].

**Figure 10 materials-14-06509-f010:**
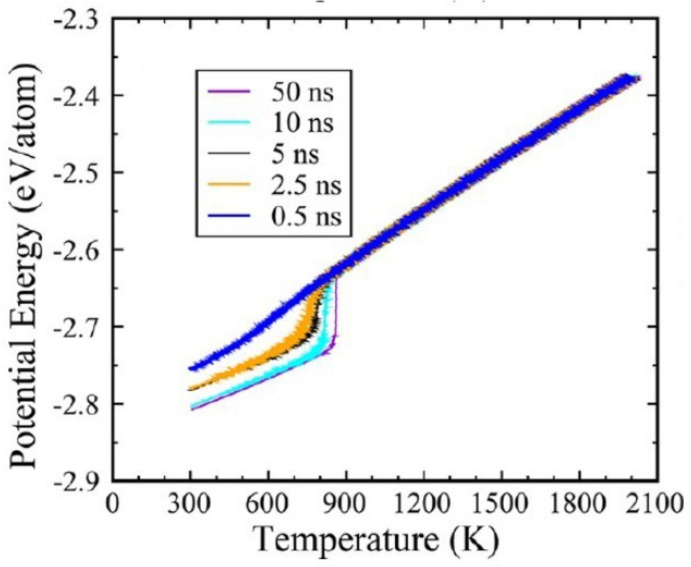
Cooling of a 256,000 atoms system with various cooling rates. −3.4 × 10^12^ (0.5 ns), −0.68 × 10^12^ (2.5 ns), −0.34 × 10^12^ (5 ns), −1.7 × 10^11^ (10 ns), and −0.34 × 10^11^ (50 ns) K/s. These transformation temperatures are predicted in Table 3-cooling. Reprinted with permission from [45]. Copyright 2020 American Chem. Society ACS.

**Figure 11 materials-14-06509-f011:**
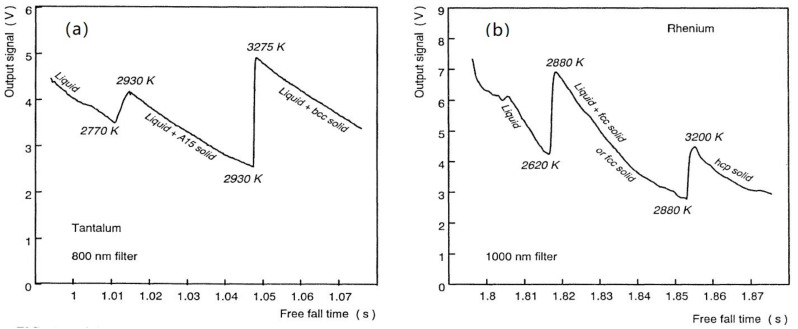
(**a**) Brightness trace during solidification of a tantalum droplet during its free fall. The two successive recalescence peaks show evidence of a double transformation phenomenon. (**b**): Brightness trace during solidification of a rhenium droplet during its free fall. The two successive recalescence peaks still show evidence of a double transformation phenomenon. Reprinted with permission from [80]. Copyright 1993 ACS.

**Figure 12 materials-14-06509-f012:**
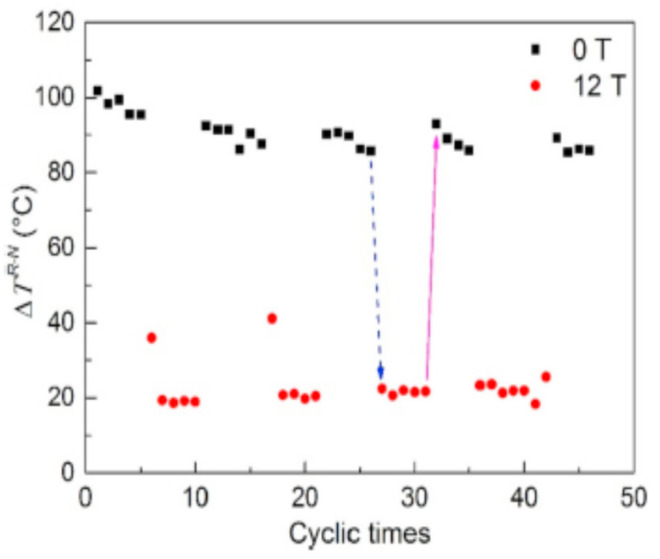
First peak of recalescence: Undercooling (∆T) of Co at different cycling times under different magnetic fields B = 0 and B = 12 Tesla. Reprinted with permission from [81]. Copyright 2019 Elsevier.

**Table 1 materials-14-06509-t001:** Lindemann coefficients δ_ls_, singular values of enthalpy coefficients −ε_ls0_ = −ε_gs0_, ∆ε_lg_ (θ_0m_), −1.25ε_ls0_, ∆ε_lg_ (θ_g_), being fractions of melting enthalpy of liquid elements. All elements, in the same column of Figure 1, have identical singular enthalpy coefficients. T_m_, T_g_ and T_D_ are in Kelvin.

	δ_ls_	ε_gs0_	∆ε_lg_ (θ_0m_)	1.25 ε_gs0_	∆ε_lg_ (θ_g_)	θ_g_	T_m_	T_g_	T_D_
Li	0.139	−0.29732	−0.16518	−0.37165	−0.13544	−0.60368	464	184	581
Ta	0.136	−0.29050	−0.16139	−0.36312	−0.13305	−0.60531	3288	1298	4109
Re	0.12	−0.25440	−0.14133	−0.31800	−0.11981	−0.61382	3458	1335	4237
Zr	0.119	−0.25216	−0.14009	−0.31520	−0.11896	−0.61433	2125	820	2600
Mg	0.113	−0.23877	−0.13265	−0.29846	−0.11377	−0.61741	923	353	1121
Ni	0.111	−0.23432	−0.13018	−0.29290	−0.11202	−0.61843	1728	659	2092
Ag	0.108	−0.22766	−0.12648	−0.28458	−0.10937	−0.61994	1235	469	1489
As	0.095	−0.19903	−0.11057	−0.24878	−0.08824	−0.62635	1090	407	1290
Fe	0.084	−0.17506	−0.09725	−0.21882	−0.08729	−0.63159	1811	667	2109
Zn	0.08	−0.16640	−0.09244	−0.20800	−0.08346	−0.63345	693	254	801
Co	0.07	−0.14490	−0.08050	−0.18113	−0.07368	−0.63780	1768	640	2013

**Table 2 materials-14-06509-t002:** Examples of second melting temperatures T_n+_ and nucleation temperatures T_nG_ of glacial phases of Tantalum, Rhenium, Zirconium, Silver, Iron, Copper and Cobalt above T_m_**.** For θ_g_ = θ_n+_ equal to singular values of enthalpy coefficients; ε_gs0_ deduced from Equation (8) with ∆ε = 0; θ_nG_ the reduced nucleation temperature of glacial phase leading to ∆ε_lg_ (θ_nG_) = ε_ls_ (θ_nG_) − ε_gs_ (θ_nG_) = −θ_n+_; ε_ls_ (θ_nG_) and ε_gs_ (θ_nG_) deduced from Equations (4) and (5); temperatures T_n+_ in Kelvin; ratios T_n+_/T_m_ and T_nG_/T_m_.

	θ_g_ = θ_n+_	ε_gs0_	θ_nG_	T_nG_	ε_ls_ (θ_nG_)	ε_gs_ (θ_nG_)	∆ε_lg_ (θ_nG_)	θ_n+_ (θ_nG_)	T_n+_	T_n+_/T_m_	T_nG_/T_m_
Ta	0.29050	3.13616	0.27222	4183	2.61326	2.90376	−0.29050	0.29050	4243	1.29050	1.27222
Re	0.31800	3.28633	0.27823	4420	2.71392	3.03193	−0.31800	0.31800	4558	1.318	1.27823
Zr	0.11896	2.39071	−0.19952	1701	2.17658	2.29554	−0.11896	0.11896	2378	1.11896	0.80048
Ag	0.12648	2.41812	−0.20456	982	2.19045	2.31694	−0.12648	0.12648	1391	1.12648	0.79544
Fe	0.21882	2.79005	−0.25049	1357	2.39618	2.61500	−0.21882	0.21882	2207	1.21866	0.74951
Cu	0.35413	3.50151	0.28445	1744	2.86407	3.21820	−0.35413	0.35413	1839	1.35413	1.28445
Co	0.0805	2.25612	−0.16895	1469	2.11122	2.19172	−0.0805	0.0805	1910	1.0805	0.83105
Max	0.47	4.37685	0.29310	-	3.53084	4.00084	−0.47001	0.47001	-	1.47000	1.29310

**Table 3 materials-14-06509-t003:** Nucleation temperatures T_nG_ of Ag glacial phases below T_m_. T_g_ and θ_g_ for various values of heating and cooling rates R in K/s; ε_gs0_ deduced from Equation (8) with ∆ε = 0; θ_nG_ the reduced nucleation temperature of glacial phase leading to singular values ∆ε_lg_ (θ_nG_) = ε_ls_ (θ_nG_)−ε_gs_ (θ_nG_) = −θ_n+_; ε_ls_ (θ_nG_) and ε_gs_ (θ_nG_) deduced from Equations (4) and (5); temperatures T_nG_; R (K/S) and LnR introduced in Equation (10) to determine T_g_ (K).

1	2	3	4	5	6	7	8	9	10	11
2	T_g_ (K)	θ_g_	ε_gs0_	θ_nG_	ε_ls_ (θ_nG_)	ε_gs_ (θ_nG_)	∆ε_lg_ (θ_nG_)	T_nG_ (K)	R (K/s)	LnR
Heating										
3	1200	−0.02826	0.88766	−0.33762	0.66000	0.78648	−0.12648	818	2.27 × 10^12^	28.45
4	1110.6	−0.10066	1.71541	−0.32584	1.30562	1.53328	−0.22766	832.5	1.7 × 10^12^	28.162
5	1023	−0.17154	1.53040	−0.25713	1.30274	1.42922	−0.12648	917	1.18 × 10^12^	27.797
Cooling										
6	925	−0.25095	1.33097	−0.41358	0.81873	1.10331	−0.28457	724	−6.80 × 10^11^	27.245
7	839	−0.32059	1.15715	−0.39673	0.74736	0.97502	−0.22766	745	−3.40 × 10^11^	26.55
8	785.6	−0.36384	1.04711	−0.34755	0.76252	0.92063	−0.15810	806	−1.88 × 10^11^	25.959
9	686.4	−0.44417	0.83156	−0.32438	0.63468	0.74406	−0.10937	834	−3.40 × 10^10^	24.249

**Table 4 materials-14-06509-t004:** Application of NCHN model to Ta, Zr, Ni, Ag, Cu, Fe and Co. θ_g_ and T_g_ for various heating and cooling rates R in K/s; ε_gs0_ deduced from Equation (8) with ∆ε = 0; θ_nG_ the reduced nucleation temperature of glacial phase leading to singular values ∆ε_lg_ (θ_nG_) = ε_ls_ (θ_nG_) − ε_gs_ (θ_nG_) = −θ_n+_; ε_ls_ (θ_nG_) and ε_gs_ (θ_nG_) deduced from Equations (4) and (5); temperatures T_nG_; T_n+_ = (1 + ∆ε_lg_ (θ_nG_)) × T_m_.

	δ_ls_	θ_g_	T_g_	ε_gs0_	θ_nG_	T_nG_	ε_ls_ (θ_nG_)	ε_gs_ (θ_nG_)	∆ε_lg_ (θ_nG_)	θ_n+_	T_n+_	T_m_	Ref.
Ta													
1	0.136	0.00000	3288	2.00000	−0.23070	2529	1.76050	1.89356	−0.13306	0.13305	3725	3288	
2	0.136	0.00000	3288	2.00000	−0.25408	2453	1.70950	1.87089	−0.16139	0.16139	3819	3288	
3	0.136	−0.49817	1650	0.67235	−0.39788	1980	0.43286	0.56591	−0.13305	0.13305	3725	3288	[84]
4	0.136	−0.49817	1650	0.67235	−0.4382	1847	0.38187	0.54325	−0.16139	0.16139	3819	3288	[84]
Zr													
5	0.119	0.00000	2125	2.00000	−0.21814	1661	1.78587	1.90483	−0.11896	0.11896	2378	2125	
6	0.119	0.00000	2125	2.00000	−0.23672	1622	1.74784	1.88793	−0.14009	0.14009	2423	2125	
7	0.119	−0.52941	1000	0.57212	−0.40785	1258	−0.35799	−0.47695	−0.11896	0.11896	2378	2125	[79]
8	0.119	−0.52941	1000	0.57212	−0.44259	1184	0.31996	0.46005	−0.14009	0.14009	2423	2125	[79]
9	0.119	−0.58118	890	0.38727	−0.49573	1072	0.17313	0.29209	−0.11896	0.11896	2378	2125	[79]
Ni													
10	0.111	0.00000	1728	2.00000	−0.21168	1362	1.79836	1.91038	−0.11202	0.11202	1922	1728	
11	0.111	0.00000	1728	2.00000	−0.22819	1334	1.76568	1.89586	−0.13018	0.13018	1953	1728	
12	0.111	−0.33449	1150	1.12207	−0.28261	1240	0.92043	1.03245	−0.11202	0.11202	1922	1728	[85]
13	0.111	−0.4618	930	0.7812	−0.3387	1143	0.57956	0.69158	−0.11202	0.11202	1922	1728	[86]
Ag													
14	0.108	0.00000	1234.9	2.00000	−0.20916	977	1.80313	1.91250	−0.10937	0.10937	1370	1234.9	[45]
15	0.108	0.00000	1234.9	2.00000	−0.22493	957	1.77233	1.89881	−0.12648	0.12648	1391	1234.9	[45]
16	0.108	0.00000	1234.9	2.00000	−0.30177	862	1.59021	1.81787	−0.22766	0.15810	1430	1234.9	
Cu													
17	0.108	0.00000	1358	2.00000	−0.30177	948	1.59021	1.81787	−0.22766	0.22766	1667	1358	[78]
18	0.108	−0.41090	800	0.92317	−0.33107	908	0.6955	0.82198	−0.12648	0.12648	1530	1358	[78]
Fe													
19	0.084	0.00000	1811	2.00000	−0.29585	1275	1.60613	1.82495	−0.21882	0.21882	2207	1811	[78]
20	0.084	−0.42736	1000	0.82123	−0.46169	975	0.42736	0.64618	−0.21882	0.21882	2207	1811	[78]
Co													
21	0.07	0	1768	2	−0.17948	1451	1.85504	1.93557	−0.0805	0.0805	1910	1768	[81]
22	0.07	−0.20136	1412	1.45491	−0.21039	1396	1.31001	1.39051	−0.0805	0.0805	1910	1769	[81]

**Table 5 materials-14-06509-t005:** Thermodynamic data for configurons in metals. T_m_ the melting temperature in Kelvin; T_n+_/T_m_ the second melting temperature divided by T_m_; S_d_ the configuron entropy in units of R = 8.34 J; H_d_ the configuron enthalpy in KJ/mol; S/S_m_ = H/H_m_ the ratio of the second melting entropy and S_m_ equal to the ratio of the second melting enthalpy and H_m_.

Metal	T_m_, K	T_n+_/T_m_	S_d_ (in Units of R)	H_d_, KJ/mol	S/S_m_ = H/H_m_
Co	1768	1.0805	44.83	686.60	0.925
Zr	2125	1.11896	30.90	578.32	0.894
Ag	1235	1.12648	29.16	318.25	0.888
Fe	1811	1.21866	17.60	292.03	0.821
Ta	3288	1.29050	13.68	422.60	0.775
Re	3458	1.318	12.64	414.67	0.759
Cu	1358	1.35413	11.53	140.19	0.738

## Data Availability

Data supporting reported results are available from the authors.

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
