# Peer review of "Prediction of Second Melting Temperatures Already Observed in Pure Elements by Molecular Dynamics Simulations"

_materials, 2021, doi:10.3390/ma14216509_

Round 1
Reviewer 1 Report
1. To what extent will the MD calculation be comparable with experiment?
2. Is it possible to apply this approach to other metals as well?
3. What is the applicability of the method to multicomponent alloys?
Author Response
The title of our revised paper is now:
“Prediction of Second Melting Temperatures Already Observed in Pure Elements by Molecular Dynamics Simulations”
All corrections are colored in yellow.
The English language is improved. The MD simulations are attributed to specialists cited in several references and reproduced figures (figures 5,6,8,9).
You raised 3 questions
- To what extent will the MD calculation be comparable with experiment?
Is it possible to apply this approach to other metals as well?
3. What is the applicability of the method to multicomponent alloys?
Answers
1-All cited simulations agree with the predictions of our model. There is only one experiment devoted to the measurement of Tg of tantalum [84] that we use to replace simulations in our model.
Reviewer 2 Report
The authors reported in this manuscript their molecular dynamics simulation on the melting of several metallic elements, with an emphasis on detecting the 2nd melting temperature of these metals. The topic is of interest, the manuscript is however poorly written, the reviewer find it difficult to recommend its publication.
1) "Liquid elements" used in the title is inappropriate.
2) Many symbols used in the equations are not explained/defined; the physical meaning and source of the numbers appeared in the equations are unclear to the audience, either.
3) Some terms used, such as "glacial phase", are not common sense, an concise definition would be of necessity.
Author Response
You estimated that the title of our paper was inadapted. It has been changed
“Prediction of Second Melting Temperatures Already Observed in Pure Elements by Molecular Dynamics Simulations”
All corrections are colored in yellow
The English language is improved and symbols definition are reexamined as you asked. The MD simulations are attributed to specialists cited in several references and reproduced figures (figures 5,6,8,9).
The term ‘glacial phase’ is explained page 3:
, the formation of stable and ultrastable glasses by vapor deposition and glacial phases by heating or annealing glass-forming melts above Tg induces new liquid states having higher glass transition temperatures.
Many thanks for your help
Reviewer 3 Report
Comprehensive report and result on Prediction of Second Melting Temperatures Observed in 2 Liquid Elements by Molecular Dynamics Simulations is well presented. A lot of reviews from other existing works were also shown in order to validate the results of molecular dynamics simulation based on NCHM model especially when predicting the various thermodynamics 2nd transitions in the melting state of selected periodic elements.
It will be nice if the following are resolved:
(1) Can you kindly include in the conclusion the trend of result of thermodynamic transitions with respect to the elements in the period Table?
(2) Since the work is mostly computational, can you kindly show briefly the set up for the molecular dynamics simulation especially the best parameters used to achieve the results?
(3) How easy it is to correlate the Linderman's criterion to the real life experimental results? Is there is any draw back or it is perfectly reliable?
Author Response
The title of our revised paper is now:
“Prediction of Second Melting Temperatures Already Observed in Pure Elements by Molecular Dynamics Simulations”
All corrections are colored in yellow
The English language is improved.
You raised 3 questions
1) Can you kindly include in the conclusion the trend of result of thermodynamic transitions with respect to the elements in the period Table
(2) Since the work is mostly computational, can you kindly show briefly the set up for the molecular dynamics simulation especially the best parameters used to achieve the results?
(3) How easy it is to correlate the Linderman's criterion to the real-life experimental results? Is there is any draw back or it is perfectly reliable?
Answers
1- The NCHN model agrees with existing MD simulations using recent values of the Lindemann coefficient of elements presented in the columns of Table 1 to predict the value of Tg. Consequently, this table and the universal law could be used to control MD simulations of other elements.
.
2- The MD simulations are attributed to specialists cited in several references and reproduced figures (figures 5,6,8,9). The set up for simulations were defined by the cited authors.
3- All cited simulations agree with the predictions of our model. There is only one experiment devoted to the measurement of Tg of tantalum [84] that we use to replace simulations in our model. The Lindemann coefficients must be known to predict Tg and the singular enthalpy values in pure elements.
For multicomponent systems, the glass transition at Tg is much higher. The NCHN model is adapted to fragile and strong liquid states because els0 of Liquid 1 in eq. 1 is no longer equal to egs0 of liquid 2 in eq. 2. The NCHN and configuron models successfully predict liquid-glass and liquid-liquid transformations in all multicomponent glass-forming melts. These predictions are possible when the glass transition temperature Tg and the melting temperatures Tm are sufficiently precise to be able to determine the singular values of Phase 3 enthalpy. The universal law relating the nucleation temperatures TnG of glacial phases and the second melting temperatures Tn+ in multicomponent alloys could be the same than that of pure elements.
Many thanks for your comments improving our paper
Round 2
Reviewer 2 Report
The manuscript has been revised to address the concerns of the reviewers, it could be recommend for publication.
The following points are suggested to the authors for consideration:
1) Line 37-38 are not quite understandable to the audience.
2) Line 43, Hm was used far before its meaning is explained.
3) Line 150 and 154, numbers used in equations are not encouraged.
Author Response
Answer to reviewer 2
Corrections are made in green.
I agree that Lines 37-38 were not clear
Line 43* : Hm is the melting heat of crystals
Lines 150-154
The presentation of equations was not clear.
Many thanks for your help
Robert Tournier